# Packing: Towards 2x NLP BERT Acceleration

**Anonymous**

## Abstract

We find that at sequence length 512 padding tokens represent in excess of 50% of the Wikipedia dataset used for pretraining BERT (Bidirectional Encoder Representations from Transformers). Therefore by removing all padding we achieve a 2x speed-up in terms of sequences/sec. To exploit this characteristic of the dataset, we develop and contrast two deterministic packing algorithms. Both algorithms rely on the assumption that sequences are interchangeable and therefore packing can be performed on the histogram of sequence lengths, rather than per sample. This transformation of the problem leads to algorithms which are fast and have linear complexity in dataset size. The shortest-pack-first histogram-packing (SPFHP) algorithm determines the packing order for the Wikipedia dataset of over 16M sequences in 0.02 seconds. The non-negative least-squares histogram-packing (NNLSHP) algorithm converges in 28.4 seconds but produces solutions which are more depth efficient, managing to get near optimal packing by combining a maximum of 3 sequences in one sample. Using the dataset with multiple sequences per sample requires additional masking in the attention layer and a modification of the MLM loss function. We demonstrate that both of these changes are straightforward to implement and have relatively little impact on the achievable performance gain on modern hardware. Finally, we pretrain BERT-Large using the packed dataset, demonstrating no loss of convergence and the desired 2x speed-up.

## 1 Introduction

Since its introduction in 2019, BERT [5] has been the backbone driving the most exciting advances in Natural Language Processing (NLP). Pre-training BERT from scratch requires substantial computational resources which may be out of reach for researchers and industry professionals. To some extent this has been addressed by the public release of pre-trained models of different sizes and depths [20]. Available sizes range from *tiny* (2 layers with hidden size 128) to *large* (24 layers with hidden size 1024)[6, 5]. The introduction of ALBERT [14] further improved the accessibility of larger models. However, the dependence on pre-trained models limits the ability of researchers to explore new backbone architectures. Furthermore, it limits the extent to which practitioners in industry can leverage internal datasets and adapt the model to their particular needs. Hence, any approach that speeds up the pre-training process is desirable from an economical as well as environmental perspective.

In this paper, we present some methods to enable researchers to accelerate the pre-training of BERT by as much as 2x. The de-facto pre-training dataset Wikipedia, as well as many other NLP datasets, show a positively skewed distribution of sequence lengths. We show that padding tokens (wasted compute) represent 50% of all tokens of the Wikipedia pre-training dataset at sequence length 512. Overall, the sample lengths range between 5 tokens up to 512 (see Figure 1). Samples of length 512 represent only 23.5% of the dataset.

While processing the padding tokens wastes compute, it is the most standard approach for leveraging modern massively-parallel compute especially on GPUs. These are most efficient when applying the same operation to each sequence in a batch. By padding all sequences to the same maximum sequence length, they can easily be batched. The most obvious way to reduce the extent of padding

in the dataset is to group samples by size before batching, i.e., process the shorter samples together and longer samples together. Typically such an approach would still involve padding but less than if padding all sequences to the same maximum length. For example BERT [5] is pre-trained in two phases, where the first phase uses sequence length 128 for 900K steps and the second phase uses sequence length 512 for 100K steps. However even by splitting the training in this way, the wasted compute due to padding is approximately 20% (see Figure 1)). Another example of this approach is Faster Transformer [18] which groups samples of similar size together in one batch and fills up with padding only to the maximum length in this batch.

More advanced approaches for reducing the padding overhead rely on custom computational kernels. Loosely these are referred to as **"un-padding"** approaches. In Effective Transformer [4], the input batch is provided as a padded matrix but padding values are dynamically removed and restored during different calculation stages. While un-padding implementations are highly sophisticated and are able to completely circumvent the processing of padding tokens, they introduce a significant overhead due to the multiple GPU kernel launches (i.e. one kernel per sequence rather than one kernel per batch). Additionally the time to process each batch will fluctuate depending on the sequence lengths in each batch i.e. batches with only shorter sequences will typically be processed faster. When working with more than one accelerator, this variability in throughput results in all devices in the cluster waiting for the device with the most compute intensive batch to finish processing. As such, un-padding approaches are not appropriate for deployment on large clusters.

The **"packing"** based approach introduced in this paper offers significant advantages over un-padding approaches. Firstly, packing is implemented directly at the framework level and requires no additional custom kernel implementations. Secondly, the processing time for each batch is independent of the content of the batch, allowing the packing based approach to maintain the same speed-up whether running on a single device or thousands. Third, each batch now contains a consistent number of real tokens.

While we demonstrate the effectiveness of packing specifically on the Wikipedia dataset, packing SQUaD [19] or GLUE datasets [22, 23] for BERT also leads to significant speed-ups (some in excess of 9x) [1] (sections A and B). The effectiveness of packing is a result of both the length distribution of the documents in the source datasets as well as the different text preprocessing steps for BERT [7]. The use of bi-directional self-attention in BERT implies that the input sequences should contain complete sentences. If a sentence is abruptly cut short, the hidden state on other (preceding) tokens in the sequence will be affected. Language models with causal attention (only attending to previous tokens in the input) do not have this issue. For such models, if a sequence is cut short at an arbitrary token, the other tokens (which occur earlier in the sequence) will not be affected. This ability to cut sequences arbitrarily completely trivializes the packing problem. For instance, GPT-3 [3] is trained with a maximum sequence length of 2048 where a single sequence may contain multiple segments separated by a special end of segment token. The last segment in each sequence is simply cut to meet the sequence length requirement. In the interest of computational efficiency GPT-3 does not mask the attention between different segments in a sequence. In contrast, the packing approach presented in this paper introduces a mask in the attention layer (see Section 3.2) to prevent cross-contamination between examples in a pack. This ensures that the characteristics of the original dataset and model are matched as closely as possible.

In summary, the contributions of the paper are as follows. In Section 2, we produce histograms of the Wikipedia pre-training dataset showing the high percentage of padding tokens. We present two new deterministic packing algorithms which easily pack datasets with millions of sequences in a matter of seconds (or less). We empirically show that the proposed packing algorithms produce a nearly-optimal packing scheme for Wikipedia pre-training dataset. We show how to compute the per-sequence loss by inexpensively un-packing the loss. We provide code for building an attention mask which prevents attention between tokens of different sequences in the pack. We demonstrate that the convergence of the BERT large model on the packed dataset is equivalent to that on the un-packed dataset. We show that with the packed dataset, we are able to achieve a nearly 2x throughput increase on the Wikipedia sequence length 512 pre-training dataset.

## 2 Wikipedia BERT pre-training dataset

BERT is pre-trained using masked-language modelling and next-sentence prediction on a large corpus of Wikipedia articles [5]. Each sequence is composed of one <CLS> token followed by the first part of sentences, followed by a <SEP> token, and then finally the second part of sentences. Because parts are created in sentence-level increments there is no token-level control of sequence length. Together with already short parts, empirically, this leads to significant levels of padding, especially for longer maximum sequence lengths (see Figure 1). At sequence length 128 (commonly used in phase 1 of pre-training) the theoretical speed-up is around 1.2, at sequence length 384 this increases to 1.7, and finally at sequence length 512 (commonly used for phase 2 of pre-training) it is 2.0. Despite the widespread use of the Wikipedia dataset for pre-training BERT such histograms have, to the best of our knowledge, not been published previously. This has perhaps lead to the underestimation of the speed-up opportunity available. To put things into perspective, the sequence length 512 dataset contains 8.33 billion tokens, of which 4.17 billion are padding tokens.

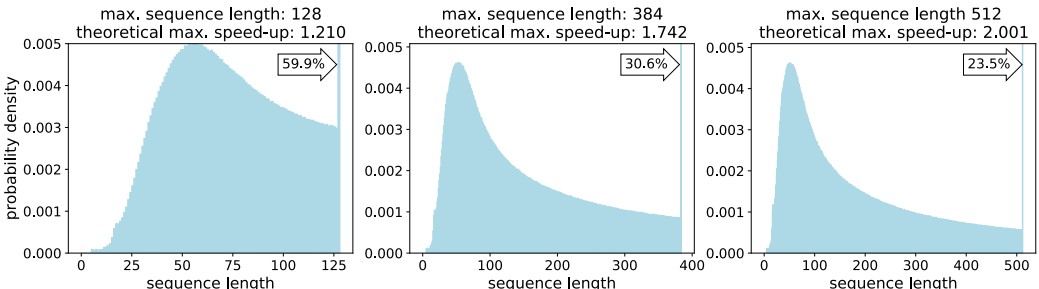

Figure 1: Wikipedia BERT pre-training dataset sequence length histograms (token count excluding padding) for different maximum sequence lengths. Based on the Wikipedia article dump from October 1st 2020. The theoretical speed-up relates to not using any padding tokens and not having any overhead from processing the different lengths.

## 3 Methods

Our approach consists of three distinct components. Firstly, we pack the data efficiently during pre-processing to make full use of the sequence length (Sections 3.1.1 and 3.1.2, see also [1] Section D). Secondly, we adapt the self-attention mask to prevent the model from attending between different sequences in the same pack (Section 3.2). Other components of the model, such as the feed-forward layer [21], operate on a per-token basis and do not require any modification. Thirdly, we compute the loss and accuracy on a per-sequence basis to match the canonical BERT implementation (Section 3.3). This is achieved by unpacking the per-pack loss at the framework level, without the use of custom kernels. Additionally, we provide suggestions for hyperparameter adjustment (Section 3.4) that lead to analogous convergence behavior between the packed and un-packed BERT implementations.

### 3.1 Packing algorithms

The problem of optimally concatenating multiple sequences of different length until a maximum combined length is reached can be directly framed as a bin-packing or stock cutting problem. Since an exact solution is strongly NP-complete [13], we propose two new heuristic algorithms that are tailored to this particular instance. A detailed introduction to packing is provided in [1](Section D).

### 3.1.1 Shortest-pack-first histogram-packing (SPFHP)

Shortest-pack-first histogram-packing (SPFHP) consists of three main components. First, the packing algorithm works on the bins in the sequence length histogram (with bin size 1) rather than the individual samples. Second, we operate on the sorted data from longest to shortest sequences. This comes basically for free due to the use of histograms. Third, we apply the worst-fit algorithm [11, 26]

onto this histogram, where the currently observed sample goes to the **pack**[1] that has the most space left to reach maximum packing depth ("shortest-pack-first"). If the sample does not fit, a new pack is created. A variant is to limit the **packing depth**, in other words the maximum number of sequences that are allowed in a pack. Therefore, we only extend an existing pack if it is not already at maximum packing depth. The detailed code for the algorithm is provided in [1] (listing 3).

### 3.1.2 Non-negative least squares histogram-packing (NNLSHP)

The proposed NNLSHP algorithm is based on re-stating the packing problem as a (weighted) non-negative least squares problem (NNLS) [2] of the form $wAx = wb$ where $x \geq 0$. The vector $b$ is the histogram containing the counts of all the sequence lengths in the dataset. Next, we define the $A$ matrix (the "packing matrix") by first generating a list of all possible sequence length combinations ("strategies") that add up exactly to the maximum sequence length. We focus specifically on strategies that consist of at most 3 sequences per pack (independent of $b$) and encode each strategy as a column of the sparse matrix $A$. For example, a strategy consisting of the sequence length 128, 128, and 256 in represented a column vector that has the value 2 at the 128th row, the value 1 at the 256th row, and zero at all other rows. The variable $x$ describes the *non-negative* repetition count for each strategy. So a 24 in the $i$th row of $x$ means that the strategy represented by the $i$th column of $A$ should repeat 24 times. Moreover, in the un-weighted setting, $Ax = b$ states that we would like to "mix" the pre-defined strategies (columns of $A$) such that the number of samples matches the histogram $b$, and where each strategy is used $x \geq 0$ times. We use the residual weight $w$ to control the penalization of the $Ax - b$ residual on different sequence lengths (different rows of $b$). Heuristically, we set the weight of 0.09 for all sequences of length 8 or smaller because they are considered acceptable padding sequences. All other sequence lengths get weight 1. After solving $wAx = wb$ for $x \geq 0$ using an off-the-shelf solver we obtain a floating point solution, which means that the repetition counts are not necessarily integers. Since we cannot use a non-natural number of strategies, we round the solution $\hat{x}$ to the nearest integer. The error introduced by this rounding is found to be negligible. Further details are provided in [1](Section D.4).

## 3.2 Attention masking for packed sequences

To maintain an implementation that is consistent with the un-packed version, we need to be able to prevent attention between tokens in the pack which belong to different sequences. Other implementations use custom attention kernels which reconstruct padding. Instead, we propose directly masking the attention matrix with a block-diagonal mask to be applied before the attention. This is straightforward to implement in modern frameworks (see Figure 2). Naturally, there is a cost to both the mask construction and applying it to the attention matrix (see Table 1, Section 4.1).

```
1  mask = np.array([[1, 1, 1, 2, 2]])   # input
2  zero_one_mask = tf.equal(mask, mask.T)   # 0, 1 mask
3  # for use with softmax:
4  softmax_mask = tf.where(zero_one_mask, 0, -1000)
```

$$\begin{pmatrix} 1 & 1 & 1 & 0 & 0 \\ 1 & 1 & 1 & 0 & 0 \\ 1 & 1 & 1 & 0 & 0 \\ 0 & 0 & 0 & 1 & 1 \\ 0 & 0 & 0 & 1 & 1 \end{pmatrix}$$

Figure 2: Attention mask code sample [left] and example zero-one mask [right].

## 3.3 Calculating per-sequence loss and accuracy

Canonical implementations of BERT compute the cross-entropy loss for the masked language model on a per-sequence basis. Simply feeding packs of sequences to the same implementation of cross-entropy would consequently result in per-pack weighting of the loss. In other words, the overall loss on the micro-batch would sum-up the losses on the individual packs, rather than individual sequences. As a result the packed BERT model would converge to a different optimum. For instance, a pack of a single sequence would contribute to the loss to the same extent as a pack of three sequences. In other words, the long sequence (single per pack) is given the same weight as the three shorter sequences in the pack of three. Empirically, a degradation of masked-language modelling accuracy on shorter sequences is indeed observed when not modifying the loss to account for packing.

---

[1] We avoid the ambiguous terms "bin" and "sample/sequence" and us "pack" instead for the entity that the document parts get aggregated in during packing. We use bin for the histogram and sequence could be a document part.

To recover the per-sequence averaging behavior of the canonical un-packed BERT implementation, it is not sufficient to simply weight the loss (accuracy) on each pack by the number of sequences it contains, because the sequences in the pack have different lengths and therefore should not use the same weight.

To implement per-sequence loss, we effectively "unpack" the incoming logits and labels by working with the per-token loss. We compute the loss on all tokens belonging to the first sequence, then all tokens belonging to the second sequence, and so on. However, rather than looping through the sequences index in this way, we compute on all indexes in parallel. This minimizes the latency overhead of un-packing the loss calculation. We use the "masked lm weight" [6] input tensor to represent which sequence a given masked token belongs to (0, 1, 2 and so on). This is consistent with the canonical BERT implementation where this input takes a value of either 1 (belonging to the sequence) or 0 (belonging to padding) as detailed in Listing 1. The same methodology can be applied to the next-sentence prediction loss and accuracy.

Listing 1: Loss calculation

```
1  # The number of sequences in each batch may vary
2  sequences_in_batch = tf.reduce_sum(tf.reduce_max(masked_lm_weight, -1))
3  sequences_in_batch = tf.cast(sequences_in_batch, tf.float32)
4  # Create the 0/1 mask that will be used to un-packed sequences
5  masked_lm_weight = tf.reshape(masked_lm_weight, [B, 1, -1])
6  sequence_selection = tf.reshape(tf.range(1, max_sequences_per_pack + 1), [1, -1, 1])
7  sequence_selection = tf.cast(masked_lm_weight == sequence_selection, tf.float32)
8  # Apply the mask to un-pack the loss per sequence
9  nll_per_token = tf.reshape(nll_per_token, [B, 1, -1])
10 nll_per_sequence = sequence_selection * nll_per_token
11 # Normalize the per-sequence loss by the number of mlm-tokens in the sequence (as is standard)
12 attempted = tf.reduce_sum(sequence_selection, -1, keepdims=True)
13 attempted = attempted + tf.cast(attempted == 0, tf.float32)  # prevent NaNs when dividing by attempted
14 nll_per_sequence = nll_per_sequence/attempted
15 # Average per-batch loss (so contributions from different batches are comparable)
16 lm_loss = tf.reduce_sum(nll_per_sequence)/sequences_in_batch
```

### 3.4 Hyperparameter adjustment

In terms of convergence behavior, the primary consequence of packing is an increase in the effective batch size (with respect to sequences and tokens) with some variation over different iterations. For instance, if each pack on average contains two sequences, the batch size (per optimization step) is effectively doubled on average. While one could subsequently reduce the computational batch size by the packing factor (average number of sequences per pack) and keep using the same hyperparameters, this is typically not desirable as it might imply under-utilizing the memory/compute.

Instead, we propose an approximate heuristic for updating the decay parameters of the LAMB optimizer [25]. For a packed dataset with a packing factor $p$, we update the decay parameters as: $\beta_1 := \beta_1^p$, $\beta_2 := \beta_2^p$. For $p = 2$, this corresponds to the exact parameters for calculating momentum and velocity, when updating with the same gradient twice [1](Section E). A common approach is to scale the learning rate with the batch size. Note however, that we take the mean gradient instead of an accumulated sum and have already a correction by the number of samples in that regard.

Since these adjustments are only heuristics the convergence of the model will be comparable but not identical. In particular, it is unlikely that simply adjusting the hyperparameters will fully undo the impact of the increased batch size. However, with these adjustments, researchers should be able to continue to use existing configurations.

## 4 Experiments

### 4.1 Bin-packing algorithm comparison

We evaluate our algorithms using the following metrics: **number of packs**, **number of all tokens**, **number of padding tokens**, **solution time of the packing algorithm** (after histogram and strategy creation), **number of strategies used**, **packing efficiency** (the fraction of non-padding tokens in the packed dataset), the **speed-up** achieved compared to not packing (depth 1), and the average number of sequences per sample (**packing factor**). For SPFHP, we analyse different (maximum) packing depth, since packing is less efficient with smaller depth and we want to get a general understanding on how the packing depth influences the processing time. For NNLSHP, we focus on packing depth 3 because it packs the data sufficiently well.

For the speed-up analysis, we focus on the intelligence processing unit (IPU) [10] (IPU-M2000, 16 accelerator chips). A GPU dynamically loads the code into the accelerator; in contrast, the IPU works with a static precompiled kernel that gets loaded onto the chip only at the beginning. While other approaches result in excessive padding or continuous changes of the code, our approach can work with the same code for the whole dataset. So in this setting the IPU architecture would especially benefit from our approach since it avoids code changes. Nevertheless, it can be applied to any implementation on GPU or TPU. For determining the speed-up, we take advantage of the precompiled kernel. Since time measurements are quite noisy, we can profile the kernel and how many cycles it takes for processing a batch. That way, we can determine the **overhead** (in cycles) from processing the additional attention masking and for unpacking the loss. Combining **overhead** and **packing factor**, we get the **speed-up** estimate. No experiment repetitions are required since the algorithms and measurements are deterministic.

The main results for the performance metric evaluation are displayed in Table 1. The processing time for SBFHP was around $0.03s$ and independent from the packing depth. We see that the overhead slightly increases with packing depth but that the benefits of packing outweigh the cost. The best speed-up is obtained with NNLSHP at depth 3. With a value of $1.913$, it is close to the theoretical upper bound of $2.001$. The results show that efficiency, packing factor, and speed-up can be viewed inter-changeably. The amount of time needed to process a sample (a pack of sequences) is barely changed relative to the un-packed implementation. The packing factor or the improvement in efficiency effectively provide an accurate estimate of the speed-up.

| pack. depth | pack. algo. | # packs [M] | efficiency (%) | pack. factor | overhead (%) | realized speed-up |
|---|---|---|---|---|---|---|
| 1 | none | 16.280 | 49.97 | 1.000 | 0.000 | 1.000 |
| 2 | SPFHP | 10.102 | 80.52 | 1.612 | 4.283 | 1.544 |
| 3 | SPFHP | 9.095 | 89.44 | 1.790 | 4.287 | 1.716 |
| 3 | NNLSHP | 8.155 | 99.75 | 1.996 | 4.287 | **1.913** |
| 4 | SPFHP | 8.659 | 93.94 | 1.880 | 4.294 | 1.803 |
| 8 | SPFHP | 8.225 | 98.90 | 1.979 | 4.481 | 1.895 |
| 16/max | SPFHP | 8.168 | 99.60 | 1.993 | 4.477 | 1.905 |

Table 1: **Key performance results of proposed packing algorithms (SPFHP and NNLSHP). Packing depth** describes the maximum number of packed sequences. Packing depth 1 is the baseline BERT implementation. Setting no limit resulted in a maximum packing depth of 16. The **number of packs** describes the length of the new packed dataset. **Efficiency** is the percentage of real tokens in the packed dataset. The **packing factor** describes the resulting potential speed-up compared to packing depth 1. With **overhead**, we denote the percentage decrease in throughput due to changes to the model to enable packing (such as the masking scheme introduced in Section 3.2). The **realized speed-up** is the combination of the speed-up due to packing (the **packing factor**) and the decrease in throughput due to the **overhead**. It is used to measure the relative speed-up in throughput and the overhead from masking and loss adjustment.

## 4.2 Learning Curves and Hyperparameter Adjustment

For depth 1 (classic BERT) and NNLSHP with depth 3, we additionally evaluate on the MLPerf 0.7 BERT pre-training benchmark [15]. Briefly, this involves training from a standard checkpoint to a masked-language model accuracy of $71.2\%$ using 3 million sequences with a maximum length of $512$ tokens (refer to [16] for details). Following this standardized benchmark supports reproduction of results even on other systems and makes sure that the reproduction effort is moderate and setup rules are clearly documented. We compare the resulting speed-up as well as the respective learning curves by evaluating the data on a held-out validation dataset. The objective of this additional evaluation is to analyse if convergence behavior is changed by the packing strategy and if the theoretical speed-up can be achieved in practice.

With packing, we effectively increase the average batch size by the packing factor ($\approx 2$). However, with a different batch size, different hyperparameters are required (see Section 3.4) and there is no mapping that will generate exact matching of results but only heuristics. In a first comparison, we use the same hyperparameters when comparing packed and unpacked training except for cutting the accumulation count by half. This way, we make sure that the batch size is constant on **average**.

In the second comparison, we evaluate our heuristics and how they compensate the difference in batch size. This setup is more desirable because it is beneficial to use the hardware to its full potential and cutting the batch size by half usually reduces throughput. In the third comparison, we compare two optimized setups.

The learning curves are displayed in Figure 3. In the first setup, we see the curves almost matching perfectly when normalizing by the numbers of samples processed. Differences can be explained by the variation of the number of sequences in the packing batch, and general noise in the training process. Especially after the initial phase, the curves show a near-identical match.

The second setup shows bigger differences since changing the batch size and hyperparameters changes the training dynamics. We observe slower convergence early on in training due to the increased batch size. This is expected. The adjustment of the learning rate actually decreases performance probably because we correct for the increased number of sequences already in the modified loss. With the adjustment of the decay parameter of LAMB, we see matching performance at the later training stages. However, it is not feasible to completely recover the early convergence behavior of the smaller batch size by adjusting the hyperparameters. For instance doubling the batch size of unpacked BERT to 3000 and adjusting the LAMB decay parameters leads to more of a slow down in convergence than when running packed BERT with a batch size of 1500 and a packing factor of 2. Overall, in practice we observe a higher acceleration than the estimated 1.913 that goes beyond 2x. We explain this with slightly better fitting hyperparameters and improved data transfer.

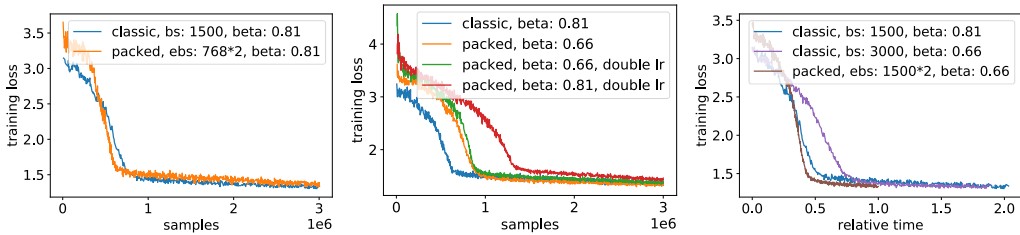

Figure 3: **Comparison of learning curves** for packed and unpacked processing with [left] same **e**ffective **b**atch **s**ize (**ebs** is batch size times packing factor), [middle] different heuristic adjustments of the hyperparameters (batch size 1500 for all runs, such that **ebs** for packed runs is $1500 * 2$), and [right] realized time-to-convergence speed-up from packing.

## 4.3 Scaling Analysis: Impact of the number of accelerators

A further advantage of packing over competing un-padding approaches is the inherent load balancing provided by packing. So called un-padding approaches rely on dynamically launching custom kernels that ignore padding. A stated advantage of such implementations is the ability to avoid computing the complete (512 x 512) attention matrix. This provides additional computational savings compared to packing, where the attention matrix is computed in its entirety and then masked. Because of these additional savings, un-padding can exceed the theoretical upper bound for speed-up from packing (2.013 on Wikipedia). As a result of the dynamic nature of the approach, the processing time with un-padding is different for each sequence in the batch, and the amount of time required to process a batch of sequences will be determined by the processing time of the longest sequence in the batch (with the sequences being processed in parallel). Furthermore, in the multiple accelerator setting the processing time on each device will vary depending on the sequences in the batch that it receives. Devices which finish early have to wait for the slowest device to finish before exchanging gradients. This load-imbalance between the devices (and inside the batch) leads to a considerable decrease in the speed-up from un-padding as the number of accelerators is increased (see Figure 4).

In contrast, packing (our approach) is inherently load-balanced. The processing time on each accelerator is independent of the content inside the batch received by the device. Any number of accelerators can therefore operate in unison without having to wait for the slowest batch to process (all per-device batches are equally fast).

To demonstrate the severity of the load-imbalance issue, we consider the scaling of an un-padding approach with a per-device batch size of 32 running on eight devices [17]. From there, we readily extrapolate the performance to both larger and smaller cluster sizes by fitting a Gumbel distribution

to the observed processing times [1] (Section F). On a single device with batch size 32 un-padding outperforms packing and exceeds the theoretical upper-bound for packing.

As the number of devices increases to two or more, the proposed packing approach outperforms the dynamic un-padding approach. On a cluster with 32 accelerators the speed-up from un-padding drops to 50% and with 2048 devices the speed-up is only 30%. In contrast, the speed-up due to packing is independent of the number of accelerators and stays at 1.913. Switching to a smaller batch size would reduce the load-imbalance issue to some extent, but would also result in under-utilization of the available memory and compute.

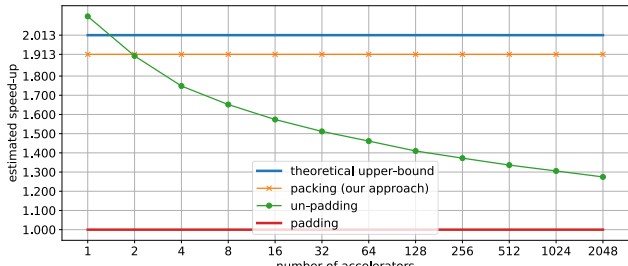

Figure 4: Comparison of the theoretical speed-up achievable as the number of accelerators is increased.

## 5 Conclusion

We showed that packing can be easily implemented without the need for any custom kernels while still providing a 2x speed-up. Additionally, we showed that any additional speed-ups resulting from dynamic un-padding approaches diminish for even moderate batch sizes or when additional accelerators are added. In contrast, packing is load-balanced and maintains the 2x throughput when scaling to large numbers of accelerators.

Furthermore, the computational overhead introduced by the attention mask and the packed per-sequence loss are small compared to the achieved acceleration. This overhead remains below 5% for all tested packing depths. The efficient packing algorithms presented in this paper enable us to pack millions of sequences in a matter of seconds. Compared to both the pre-processing time for the Wikipedia dataset and the training runtime, this overhead is negligible. Furthermore, we showed that performing packing as a pre-processing step does not significantly impact the training convergence. Our proposed hyperparameter adjustment scheme additionally helps practitioners easily modify existing validated optimizer settings for use with packed BERT. Further exploration of hyperparameter selection is left to future work.

When performing packing as a pre-processing step, the proposed NNLSHP and SPFHP methods achieve near optimal compression efficiency. In this offline setting, we are able to build a histogram of the dataset, and thus achieve linear time complexity with respect to the number of samples. This makes packing modern datasets with millions of sequences possible. In the future, it would be interesting to extend SPFHP to the online setting where a histogram of the entire dataset cannot be built. Another interesting direction is the packing of images of different sizes to help accelerate computer-vision applications. This is especially relevant given the recent advances in the use of transformer-based approaches in the computer vision domain, for example the visual transformer [24]. Masking out the self-attention within transformers is easier to implement than avoiding cross-contamination of convolutions applied to packed images. Finally, packing could potentially eliminate the need for two phase pre-training of BERT. Using short sequences in the first phase to reduce the waste from padding is no longer attractive for packed sequence BERT where the padding is essentially a negligible proportion of the tokens. Furthermore, the argument that the model should first learn short-term dependencies by training on short sequences neglects the fact that these same short-term patterns can be learned from longer sequences. In fact, longer-sequences may contain multiple short patterns, while also maintaining long-range consistency. Future work should explore training packed BERT from scratch and the impact of packing on fine-tuned performance.

## Broader Impact

We showed that when pre-training BERT on Wikipedia, the computational overhead taken to process padding tokens is roughly $50\%$. By eliminating this wasted computational time, the approach presented in this paper paves a way to halving the carbon footprint of training BERT-based models.

Furthermore, our approach circumvents the need for custom kernels, making the benefits of packing readily accessible to a broader audience of NLP practitioners. As such we are hopeful the research will have a positive impact on the NLP community, and do not see any disadvantage of using this approach.

Future work would need to investigate the applicability of packing on text produced by different cultures and in different languages. We have already shown that the speed-up resulting from using our methods does not only occur when pre-training BERT on Wikipedia but also on other datasets such as SQUaD and GLUE. Furthermore, the sentence length distribution of the original English language text shows similar characteristics. Our research leads us to believe that compressible distributions arise naturally in language tasks and beyond, for instance in DNA sequence lengths [9] and protein lengths [8]. Many such sequence modelling workloads are based on variations of the BERT/transformer architecture and would therefore easily benefit from our acceleration.

Failures in NLP can have a big impact on society; many technologies, such as Alexa, Siri, and Google Home, rely on them. Whilst any errors arising from our approach can be avoided, one potential source of error comes from the implementation. Both the attention mask and the per-sequence loss need to be modified to support packing. These changes are significantly smaller than those required by custom kernels, however they may still be time consuming to implement and debug. To help mitigate the risk of any implementation errors, we share our reference implementations of the required changes in the supplemental material [1].

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
