# OpenReview forum: "Packing: Towards 2x NLP BERT Acceleration"
_NeurIPS.cc/2021/Conference — NeurIPS 2021 Submitted_

### Official Review · Reviewer_QSu8 · 2021-07-14

**Rating:** 5
**Confidence:** 5

**Summary:**

This paper proposed two algorithms of packing (batching) variable-length sequences in training deep NLP models, SPFHP and NNLSHP. Experiment shows that these algorithms can accelerate BERT training by 2x compared with naive padding strategies and that they are also more scalable than un-padding strategies.

**Main Review:**

Padding and packing (batching) is an important detail in large-scale NLP model training, but it has long been ignored by many NLP researchers and practitioners. The topic of this paper is both innovative and highly practical. Two packing algorithms proposed by the authors are also very close to the optimal, so the author almost "solves" this problem entirely. The proposed algorithms only add little overhead to the whole training process, and they are also more scalable and flexible than un-padding methods. The paper also discussed how to tune hyperparameters given the new packing strategy to achieve good performance.

Still, I think there are still many problems in this paper. There are two major issues and two minor issues.

Surprisingly, **packing is not a problem in masked language model (BERT) pre-training**. The RoBERTa paper (https://arxiv.org/pdf/1907.11692v1.pdf) has talked about this problem in detail. Table 2 of the RoBERTa paper shows that the sentence-pair scheme used by the original BERT codebase is worse than "Full-Sentences" or "Doc-Sentences", where multiple contiguous sentences are packed into a single sequence, such that the total sequence length is at most 512 tokens. **Language pre-training never requires each sequence to be a single sentence**; instead, it can be a concatenation of an arbitrary number of contiguous sentences. Therefore, the discussions in Section 3.2 (attention masking) and Section 3.3 (calculating per-sequence loss) seem useless, since cross-contamination between contiguous sentences is not harmful but even beneficial (as is shown in the RoBERTa paper). Although the proposed strategies are perhaps useless in language pre-training, **they can still be useful in other NLP tasks**, where the training dataset is a noncontiguous set of variable-length sentences. These tasks are machine translation, sentence classification, QA tasks, and so forth.

However, in some of these tasks, **practitioners are already using packing strategies better than naive batching**. For example, the fairseq codebase (https://github.com/pytorch/fairseq/blob/f2146bdc7abf293186de9449bfa2272775e39e1d/fairseq/tasks/fairseq_task.py#L208) for machine translation uses following packing strategy:

1. Sort sentences in the training dataset w.r.t. their lengths.
2. Iterate over sentences in the sorted dataset and try to add sentences greedily into the current batch.
    - If the resulting batch has more than n_tokens (a preset hyperparameter), finish the current batch and put this sentence into a new batch. Otherwise, add the sentence to the current batch.
3. Pad all batches so that all sentences have the same length as the longest sentence in the batch. Since sentences in the same batch are contiguous sentences in the sorted dataset, they have similar lengths. Thus we will not end up adding too many padding tokens.
4. This batching strategy also prevents cross-contamination automatically (since different sentences are not even in the same sequence), and it is even easier to calculate the per-token loss.

Although this strategy is probably never published, it is widely used in many NLP projects (https://github.com/pytorch/fairseq#pre-trained-models-and-examples), especially machine translation (https://github.com/pytorch/fairseq/blob/master/examples/translation/README.md). Unfortunately, **the author of this paper ignores this extremely commonly used baseline**. Instead, the author compares their algorithms with *naive padding*. However, except for small downstream NLP tasks, naive padding is almost only used by the vanilla BERT codebase and some projects using the Huggingface *transformers* codebase, probably for compatibility with Google's TPU instead of efficiency.

Writing is one of the minor issues. The description of the algorithm and TensorFlow source codes are informative and clear. However, unlike other algorithm papers, this paper has no derivation of (sequential) time complexity, parallel time complexity, and space complexity of their algorithms.

Another minor issue is that the author fails to explain how they have chosen *the weight of 0.09 for all sequences of length 8* for NNLSHP. If these hyperparameters are manually tuned, it would be better to add a sensitivity analysis showing whether the efficiency depends on a specific choice of this hyperparameter.

To summarize, this paper chooses an interesting topic and tries to solve this problem instead of making incremental improvements. The proposed methods are efficient and scalable. However, the author tries to solve a non-existent problem in BERT pre-training, although they may still be helpful in other NLP tasks other than LM pre-training. Also, the author only compares the methods with the weakest baseline, ignoring a very commonly used strong baseline. Therefore, I think this paper needs to be reworked.

**Time Spent Reviewing:**

6

---

> ### Author Response · Authors · 2021-08-10
> **Thank you for your review**
>
> We thank the reviewer for agreeing that "the topic of this paper is both innovative and highly practical." We are sure we can address all their concerns during the review process. We especially thank the reviewer for pointing us in direction of the RoBERTa paper, which we will address in our revised paper.
>
>
>
> The BERT dataset generation strategy as well as RoBERTa both concatenate contiguous sentences. In the BERT approach, there are 2 segments of contiguous sentences. With 50% probability those two segments come from the same contiguous text, and the role of next sentence prediction is to predict if the two segments are contiguous. Our masking approach prevents contamination between different sequences within a pack, but not between sentences and segments within the same sequence. In the FULL-SENTENCE approach in RoBERTa, sentences from different documents get concatenated to get almost full batches. However, some padding remains, and we are currently exploring how much padding remains and how our packing can remove even this padding by more carefully selecting the documents. More importantly, we would like to point out that the RoBERTa paper says that avoiding cross-contamination between documents, as done by their DOC-SENTENCES strategy, slightly improves performance. So, our paper can help for this approach two-fold: either by removing the contamination in the FULL-SENTENCES approach or by applying packing to the DOC-SENTENCES approach. Note that the RoBERTa paper selects the FULL-SENTENCE approach for computational efficiency. Our approach can provide the best of both worlds and resolve the difficulties in load-balancing that were briefly mentioned in the RoBERTa paper.
>
>
>
> Creating the RoBERTa datasets (DOC-SENTENCES and FULL-SENTENCE) is quite a cumbersome process. If the reviewer could share the respective histograms, we would gladly run our histogram analysis on it together with the results for packing. We believe that by publishing our paper, more practitioners will become aware of the potential to speed up applications and processing of datasets which are not covered in the paper, such as RoBERTa.
>
>
>
> Cross-interaction between "contiguous sentences" is indeed desirable in BERT and RoBERTa. The masking introduced in section 3.2 is used to avoid cross-contamination between completely different sequences in the same pack, and not contiguous sentences. In terms of RoBERTa, the masking approach could prevent interaction between different documents as they occur in the FULL-SENTENCES approach.
>
> The loss-unpacking of Section 3.3 is mostly included to show that per-sequence loss calculation remains possible in general, such that NLP tasks which depend on it can still be accelerated. As noted in the review, the masked-LM losses and accuracy are usually calculated per-token and not per-sequence, while the NSP loss (which is calculated per-sequence) is shown in the RoBERTa paper not to be a crucial part of pretraining.
>
> We provide the code snippet to simplify the adaption of our approach across a variety of NLP tasks. However, we would be willing to move the section into the appendix.
>
>
>
> The "sorted batched padding" is analogous to the Faster Transformer approach cited in the paper. However, we realize that fairseq is just as prominent (if not more so) and will add a reference. There are 4 reasons why this approach is not a baseline for our research, and why it effectively would not be an apples-to-apples comparison.
>
> 1. Sorting data is quite uncommon in machine learning to avoid overfitting. This holds especially on larger datasets and batches.
> 2. To make optimal use of most accelerators, the batch size must be increased when the sequence length drops and load balancing challenges can occur which have to be solved individually for each hardware without any general approach.
> 3. HuggingFace and naive batching are still broadly common in practice.
> 4. Changing the batch size requires transferring different code/kernels to the accelerator for each size instead of using one setup for one size. Modern chips like the Graphcore IPU, as well as computational graph optimization frameworks like TensorFlow XLA, especially benefit from keeping the batch size constant. This was highly relevant for our recent MLPerf submission where "sorted batched padding" would have decreased performance instead of increasing it. GPU submissions did not use "sorted batched padding" either in the benchmark but unpadding which indicates that it has better performance.
>
>
>
> We did not want to make the focus of the paper about GPU/TPU/IPU comparison and MLPerf because, as agreed by the reviewer, we are providing a much more general (to some extent already known but relatively underpublished) approach that generalizes to a variety of problems, datasets, and hardware. If desired, we can emphasize these design aspects in the introduction.
>
>
>
> Are there any other references beyond the fairseq library where the "sorted batched padding" is used/described that we should additionally reference in our paper? What would be an appropriate name for the approach?
>
>
>
> For the packing algorithms, we address time and space complexity in the supplementary material. We can make the respective results more prominent in the main paper. In fact, our analysis proved the SPFHP algorithm to be highly efficient, whereas the NNLSHP algorithm scales perfectly with the number of samples but gets into space and time complexity issues with increased sequence length (squared memory and even higher order time complexity) and increased number of packed sequences (changing from squared to cubed complexity for the memory when packing up to 4 sequences together which exceeded our available memory).
>
>
>
> For "the weight of 0.09 for all sequences of length 8 for NNLSHP", we tested only a few numbers. After we reached almost perfect packing efficiency, we stopped looking into the parameter. We are already adding the sensitivity analysis to the supplementary material and will address the choice of this parameter in the paper.
>
>
>
> Thank you for pointing us in direction of the RoBERTa paper and concluding that "this paper chooses an interesting topic and tries to solve this problem instead of making incremental improvements. The proposed methods are efficient and scalable." We are certain that we can address the open issues during the review process.

---

> > ### Author Response · Authors · 2021-08-14
> > **Weighting analysis**
> >
> > We added the following sentences to the main paper:
> >
> > "*Heuristically, we set the weight of 0.09 for all sequences of length 8 or smaller because they are considered acceptable padding sequences. All other sequence lengths get weight 1. We discuss this strategy and the choice of parameters in [1](section D.4.5 and D.5). Whereas, solutions looked visually more appealing with weighting, our evaluation showed that there is only little benefit from weighting on the speedup (less than 1%).*"
> >
> > In the supplemental material, we added a full section on the weighting sensitivity analysis:
> >
> > *D.5    Discussion of residual weight choice*
> >
> > *This section discusses the choice and effect of the weighting parameters in the NNLSP packing algorithm.  To simplify the problem of selecting reasonable defaults for the residual weights, we use just two parameters to completely describe the weights: an “offset” parameter and a “weight”parameter.  Originally, all sequence length residuals are given the same weight of 1.  This results in a packing with leftover long sequences, because there are not enough short sequences to pack them with. To reduce the residual on long sequences, we could either increase the residual weight on long sequences or reduce the weight on short sequences.  We chose to reduce the weight on short sequences. Specifically, sequence lengths up to the “offset” length have a reduced “weight”. The other residual weights stay at 1.*
> >
> > *To start, we chose an offset of 8 tokens, which is the smallest power of 2 for which there are examples in the Wikipedia dataset. We decrease the weight on sequences shorter than the “offset” from 1 to 0.920 to 0.09 to see which order of magnitude is the most appropriate. On visual inspection (looking at the residual plots as in Figure 11), we found that 0.9 still left too many long sequences unpacked. So, we reduced the weight a further order of magnitude to 0.09. This seemed sufficient to encourage nearly all long sequences to pack.  While visual inspection helps in understanding how many long/short sequences are leftover, we are also interested in the impact the weights have on the overall efficiency of the packing.*
> >
> > *Without any weighting, we get 99.746359% efficiency, whereas the weighted approach results in 99.746274% efficiency. Hence, we conclude that the impact of the weights on the packing efficiency is very limited. Additionally, using an “offset” length of 4, resulted in similar numbers, for the full range of weights from 0 to 1. Using a weight of 0 for an “offset” length of 8 resulted in insignificantly higher efficiency of 99.7519%, whereas using an “offset” length of 16 reduces performance to 99.38964%. A weight of 0 implies that the residual on those lengths can be safely ignored, i.e., the packing algorithm can thus add as many short sequences as it chooses without any penalty. It is very interesting that this does not significantly impact the packing efficiency, and can even have a slightly positive impact. However, increasing the “offset” length further significantly decreases the performance with weight 0. Keeping the weight at 0.09 and increasing the length reduces performance slightly, for example with 99.53% at length 256 and 99.728% at length 16.*
> >
> > *For our Squad analysis, weighting improved the efficiency slightly from 96.94% to 97.38%. Fine tuning further with direction grid search, delivered a local optimum of 98.767% efficiency with length 64 and weight 0.002.
> > Overall the influence of different residual weights on the packing efficiency (and the acceleration factor) is less than 1%. This might differ from application to application, but it shows that we are able to use the residual weights to achieve secondary targets (like not having leftover long sequences) without significantly compromising the packing efficiency.*
> >
> > We hope this clarifies our approach and is a sufficient sensitivity analysis.

---

> > ### Author Response · Authors · 2021-09-02
> > **Addressing RoBERTa and fairseq**
> >
> > We modified parts of the introduction to discuss the sorted batching approach even further:
> >
> > *"While processing the padding tokens wastes compute, it is still the most standard approach for leveraging modern massively-parallel compute especially on GPUs. These are most efficient when applying the same operation to each sequence in a batch.  By padding all sequences to the same maximum sequence length, they can easily be batched.  We note that this naive batching is the most widely used and provided in the vanilla BERT implementation as well as the Hugging Face framework [29] and thus considered as our baseline for comparison.*
> >
> > *The most obvious way to reduce the extent of padding in the dataset is to group samples by size before batching, i.e., process the shorter samples together and longer samples together. BERT [6] is pre-trained in two phases, where the first phase uses sequence length 128 for 900K steps and the second phase uses sequence length 512 for 100K steps. However even by splitting the training in this way, the wasted compute due to padding is approximately 20% (see Figure 1). Other examples of this “sorted batching” approach are Faster Transformer [22], fairseq [23], and RoBERTa [17], which group samples of similar size together in one batch and fill up with padding only to the maximum length in this batch.  This approach can be highly efficient in cases where the dataset length is multiple orders of magnitude larger than the batch size and the number of different sequence lengths. Despite its high computational efficiency, this approach has multiple drawbacks. We outline these below and propose an alternative which maintains the high efficiency, while also circumventing the downsides. Firstly, sorting the data can reduce the overall convergence speed when the batch size is large because it violates the i.i.d. assumption on the data distribution [2,19]. Secondly, processing batches with shorter sequence lengths under-utilizes the compute compared to running the same batch size with a longer sequence length. For GPUs, a common heuristic to mitigate this effect is to adjust the batch size to keep the number of processed tokens near constant [23,17]. In general however, the relationship between the sequence length and the optimum batch size is more complex and maximizing compute utilization can require the model to be sharded differently across multiple accelerators. Avoiding this, often manual process, is important for ease of use and the portability of methods across different hardware architectures. Thirdly, modern NLP applications are optimized and compiled for fixed tensor sizes using tools such as XLA [31,9], which provides a  7x acceleration for BERT MLPerf™[18] compared to the non-XLA baseline [31]. Changing the sequence length or batch size requires re-optimization of the computational graph and recompilation of the program for the new tensor shapes. For complex models (such as BERT) this optimization and recompilation take a non-negligible amount of time. Even if one pre-compiled and cached all combinations of batch size and sequence length, the kernels would still need to be re-uploaded to the device every time the shapes change. Depending on how frequently the tensor shapes change, the overhead from switching kernels adds up. To avoid these issues, it is preferable (and common) to work with fixed tensor shapes for the entire duration of the training run."*
> >
> > We now address further differences to RoBERTa at multiple occasions in the revised paper like for example:
> >
> > *"RoBERTa [17] uses a similar packing approach as GPT-3 but with full sentences. Their ablation study shows that mixing of sentences from different documents reduces accuracy, but it is used nonetheless for load balancing reasons."*
> >
> > in the introduction and in the conclusion, we mention:
> >
> > *"Finally, whereas BERT used a two-phase pre-training, RoBERTa proposed to rather skip the first phase to obtain better performance. Hence, we can expect that the first phase can be skipped in BERT, too, making our acceleration even more beneficial. Future work should explore training packed BERT from scratch without the first phase, the impact of packing on fine-tuned performance, and improving the performance of the RoBERTa approach by avoiding contamination between non-contiguous segments from different documents."*
> >
> > Are we missing anything else?

---

> > ### Comment · Reviewer_QSu8 · 2021-09-11
> > **Thank authors for their explanations**
> >
> > Thank the authors for their explanations to address my concerns during the review process.
> >
> > About RoBERTa: indeed, FULL-SENTENCE in RoBERTa does not give the best performance. However, the performance number is similar to the best strategy DOC-SENTENCE. They are also better than SENTENCE-PAIR where there is no cross-contamination. This means that the average sequence length in RoBERTa-style language pre-training is usually close to the pre-determined maximal sequence length (512). Therefore, we need quantitative comparisons (comparing to RoBERTa instead of BERT) to show whether the packing methods described in this paper are still useful.
> >
> > About sorted batched padding (tend to overfit): I do not agree that sorted batched padding will cause overfitting for many NLP applications. Indeed, sorting the dataset will reduce the randomness from data shuffling, decreasing its regularization effect. However, this is not a big issue for sorting according to the sentence length. Since the size of the dataset (> 100k for machine translation) is usually significantly greater than the number of possible sequence lengths (512), we still have enough randomness if we do a stable sort after a random shuffle. Shuffle+Stable sort is the default behavior of Fairseq, which implemented or reproduced many state-of-the-art machine translation systems and some NLU systems too.
> >
> > About sorted batched padding (variable batch size): I agree with the author that sorted batched padding will introduce variable batch size, which will make some implementations less efficient, especially for TPU and specialized hardware. Since it is unfair and different in this sense, I think it is okay not to compare with this baseline, as long as it is mentioned in the paper.
> >
> > In the rebuttal, the author successfully addressed some of my concerns (sorted batched padding, hyperparameters, etc.) and added many explanations. But a direct comparison with RoBERTa is still missing, and the method is still mainly compared with an inefficient weak baseline (the original implementation of BERT). Therefore, I decided to increase my score on this paper from 4 to 5.

---

> > > ### Author Response · Authors · 2021-09-14
> > > **Thank reviewer for comment and adjustment**
> > >
> > > We thank the reviewer for the revised feedback. We understand and share the reviewers passion about the RoBERTa paper and the fairseq library. Indeed, the original BERT paper leaves a lot of room for exploration like provided in the RoBERTa paper and it is important to get a better understanding how to improve the BERT algorithm.
> > >
> > > We would like to clarify, that our paper is about a new acceleration technique with a lot of implications that can benefit plenty of applications far beyond RoBERTa and BERT and future work can address these explorations. For this paper, the focus was to show that our algorithm does not impact accuracy but can improve speed, which from our point of view is a crucial property for its adoption. Thus, we stick to the only NLP related industry standard benchmark which is provided in MLPerf(TM) to have a very controlled and clearly specified scenario. Given the findings, it is safe to assume that our packing strategy transfers to other approaches like RoBERTa, GPT like models, and many more in two aspects: We can either pack data and get a large speed-up or we can avoid document contamination and obtain an increased model accuracy at the price of only a little overhead.
> > >
> > > Inspired by the review, we re-read the RoBERTa part on SENTENCE-PAIR and SEGMENT-PAIR and had three crucial discoveries.
> > > First, we would like to point out that in SENTENCE-PAIR and SEGMENT-PAIR there is a 50\% chance that data from different documents lands in the same sequence, only separated by a separation token. However, the different documents are still able to attend to each other. To the best of our knowledge, the impact of this "different" kind of contamination on accuracy has not been addressed at all yet, but could be covered in future by our modified mask. We will add this observation to our future work section.
> > > Second, SENTENCE PAIR is not able to learn long range attention that goes beyond a sentence. This reduces its performance significantly (1-2\%) and makes it irrelevant for any comparison.
> > > Third, the FULL-SENTENCE has probably almost no padding tokens. In contrast, SEGMENT-PAIR "follows the original input format used in BERT"[Liu2019] and therefore similarly has 50\% non-padding tokens.
> > >
> > > The comparison in Table 2 of the RoBERTa paper uses 1M training steps with a batch size of 256 (FULL-SENTENCE) resulting in a total amount of 150 Billion tokens (1Mx256x512). However, the amount of tokens that goes into the training of SEGMENT-PAIR is 50\% less (75 Billion tokens), corresponding to effectively only 500K training steps compared to FULL-SENTENCE.
> > > On the other hand, in Table 4 of the RoBERTa paper increasing the number of training steps improves performance by around (0.5-1.7\%) which is much more than the than the observed performance difference between FULL-SENTENCE and SEGMENT-PAIR (-0.4 to +0.7\%). **Based on this data, it seem that if we train SEGMENT-PAIR twice as long to be comparable with FULL-SENTENCE, the performance gain would be bigger and SEGMENT-PAIR would be the top performing data preparation scheme.** Our paper provides the capability to run SEGMENT-PAIR at twice the speed.
> > >
> > > Addressing this discrepancy (SEGMENT-PAIR vs. FULL-SENTENCES) is a big challenge and is required before comparing to the RoBERTa algorithm. The fairseq library provides the RoBERTa model itself but we could not find the scripts that were used for all the experiments in the paper to reproduce the results and check the data generation strategies in detail. The respective required experiments are worth a publication on its own and go way beyond the scope of our submission. Thus, we have to leave the application of our packing strategies to RoBERTa for future work.
> > >
> > > To be more transparent and fair about the comparison to RoBERTa and other baselines, we address them in the state of the art and we will add additionally a paragraph to the Broader Impact section to comment on this limitation of our work.
> > >
> > > "The benefit of our algorithm is based on two assumptions: A skewed length distribution in the training dataset and a hardware setup that trains efficiently on a fixed batch size. If **efficient** training is possible, with a variable batch size approaches like FasterTransformer and the fairseq sorted batch approach will result in the same or even larger benefits (due to smaller self-attention matrices). If the dataset is generated differently like in GPT models [Brown2020] and RoBERTa (FULL-SENTENCES) [Liu2019], all sequences will be at full length and sequences cannot be concatenated and there is indeed no benefit in packing sequences.
> > > However, strategies that reach full sequence length usually combine segments from different unrelated document sources which can result in reduced performance [Liu2019]. Even in the normal BERT model, there might be this contamination between segments from different documents. Our paper introduced an approach to avoid the contamination between sequences. However, the same approach could also be applied to avoid contamination between segments and it remains future work to explore its benefits beyond BERT pretraining."

---

### Official Review · Reviewer_b88q · 2021-07-17

**Rating:** 6
**Confidence:** 5

**Summary:**

Authors found that at sequence length 512 padding tokens represent in excess of 50% of the Wikipedia dataset used for pretraining BERT (Bidirectional Encoder Representations from Transformers). Authors pre-train BERT-Large using the packed dataset,
demonstrating no loss of convergence and the desired 2x speed-up.

**Limitations And Societal Impact:**

Yes authors addressed the limitations and potential negative societal impact of their work. Failures in NLP can have a big impact on society; many technologies, such as Alexa, Siri, and Google Home, rely on them. Whilst any errors arising from our approach can be avoided, one potential source of error comes from the implementation.

**Main Review:**

Authors showed that packing can be easily implemented without the need for any custom kernels while still providing a 2x speed-up. Additionally, we showed that any additional speed-ups resulting from dynamic un-padding approaches diminish for even moderate batch sizes or when additional accelerators are added. In contrast, packing is load-balanced and maintains the 2x throughput when
scaling to large numbers of accelerators.

**Time Spent Reviewing:**

3 hrs

---

> ### Author Response · Authors · 2021-08-10
> **Thank you for your review**
>
> Thank you for the positive feedback.

---

### Official Review · Reviewer_Sn5a · 2021-07-18

**Rating:** 6
**Confidence:** 3

**Summary:**

This paper proposes a simple and effective approach to accelerate the standard BERT pretraining by packing multiple sentences to a single training sample with fixed length, in order to reduce padding tokens and fully utilize the computation. The packing algorithms itself are very efficient and the overhead introduced in training (attention mask and loss calculation) is limited (below 5%). Experiments show that with hyperparameter adjustment, the BERT pretrained with packed data can result in nearly 2x speed up. Compared to the unpadding approach with custom computational kernels, the packing approach is inherently load-balanced and thus scale better to multiple distributed accelerators. Overall, I believe this is a quite general approach that can be applied to large-scale training with data having variable lengths.

**Ethics Review Area:**

["I don’t know"]

**Limitations And Societal Impact:**

Yes, the author has addressed this issue in the paper. I think this work have positive impact since it can save the carbon footprint in training the large-scale models.


**Main Review:**

This paper with its supplementary document is general well designed with solid experiments and analysis. To me, the idea of packing data with different lengths is not new, has perhaps arguably been tried by researchers or engineers in their working on accelerating BERT training, but not published. Even in the original code of BERT, the preprocessing step considers concatenating multiple segments into one a long sequence, although a significant portion of padding tokens are still introduced.

The paper can be further improved in the following aspects:
* Section 3.1 (packing algorithms) is hard to understand, especially to the readers with little background of packing algorithm. It will be better to have a formal definition of the problem, rather only explaining in words.
* Section 3.3 (calculating per-sequence loss) has the similar problem. It is unclear to me how the losses are computed and aggregated. It would be better to have some formula and figure to present the process, rather than the code snippet.
* In the packed data, how are the positional encodings configured? The positions of multiple sentence in one sample may have influence to the model behavior.
* The proposed method are only evaluated in terms of the pretraining loss, while in practice it may be more important to evaluated the performance of the model (packed BERT) finetuned on downstream tasks. I am wondering whether there will be differences with original BERT on certain tasks.
* “Empirically, a degradation of masked-language modelling accuracy on shorter sequences is indeed observed when not modifying the loss to account for packing”, what is exactly the drop of accuracy in the experiment?
* “packing could potentially eliminate the need for two phase pre-training of BERT.” It will be better to valid this argument, since the two stages pretraining could have the effective of curriculum learning which can speed the coveragence.

Overall, I think the paper provide a simple and effective method to accelerate BERT training, although the idea could be employed by other practitioners. Some part of the paper is not easy to follow and could be improved with better formulation or graphs. Experiments are generally solid, but lack of experiments on the downstream tasks and analysis on the positional embedding.


**Time Spent Reviewing:**

6

---

> ### Author Response · Authors · 2021-08-10
> **Thank you for your review**
>
> We thank the reviewer for their positive feedback and for pointing out that the paper "is general well designed with solid experiments and analysis". We agree that "this is a quite general approach that can be applied to large-scale training with data having variable lengths." We hope that this was made sufficiently clear in the conclusion. For the final version, we will add that other derivative pretraining approaches (like RoBERTa), downstream tasks, or any other algorithm with self-attention can benefit from our technique. We therefore hope to present the paper at NeurIPS to introduce it to an audience that covers a wide range of applications with different sequence length and the latest algorithm developments. The inclusion of code snippets is also intended to aid in the adoption of packing to other applications and algorithms.
>
>
>
> We agree that packing is nothing new in itself and has been applied in another context even before 1970. We can address this further in revising the state of the art, especially by referring to GPT3, RoBERTa, and early literature, and by clarifying even further the shortcoming of the original BERT "packing" approach. The original BERT "packing" only referred to putting multiple sentences together, but this did not sufficiently alleviate the padding problem. GPT3 is packing data but since there is no bidirectional self-attention, data can just be added and there is no need for any full sentences or special adjustments. The RoBERTa paper concatenated sequences (from different documents) but did not address the problem of efficiency or how to avoid cross-contamination, even though they discovered that it slightly degrades performance.
>
>
>
> However, we do provide two main contributions that have not been addressed in the past (or hidden) and which justify a publication. Firstly, we introduce highly efficient packing strategies that scale well and provide a very good solution even for the 3-partition problem. Secondly, we provide an approach to avoid cross-contamination between packed sequences in self-attention layers and show how to un-pack the loss function. These contributions allow practitioners to side-step the need for custom kernels which can be difficult to develop/maintain and tie you down to particular hardware.
>
> To additionally address the point that packing is nothing new, we can consider rephrasing the paper title to mention "Efficient Packing".
>
>
>
> In discussions with multiple key players from major companies in machine learning, they stated that our packing approach was completely new to them. They then started exploring this approach after we talked with them about it. They found that our approach would prevent them from being locked into a hardware choice, as it removes the need for custom kernels. Hence, we doubt that there was anything hidden before we communicated it. Even if there were something hidden, it would be beneficial to engineers if our work were released to the public to enable all engineers to work with our approach and benefit from its efficiencies.
>
>
>
> All remaining issues can be addressed in a revised version. We were hesitant to elaborate too much on packing in the paper to avoid the criticism that it is a well-known topic, and we therefore moved the content into supplementary material. We would happily reinsert a more elaborate introduction into the main paper.
> In the loss code, we are just transforming the packed sequence into an unpacked structure that corresponds to the sequences, since we know the maximum number of sequences in a pack. The un-packing itself is not something that can be shown very clearly with a formula. However, we would like to add a general loss formula in that section to demonstrate more clearly what we are trying to accomplish.
>
>
>
> Thanks for bringing up the issue of positional encodings; we will add a section into the paper to show how we handle it in the implementation: We perform an embedding look-up to extract the correct position embedding vector for each token in the pack. This requires keeping an extra input which specifies the position of each token in its sequence. For instance, if you pack two sequences, one of length 2 and one of length 3, the positions you would want to pick up with the embedding look-up are [0, 1, 0, 1, 2].
>
>
>
> Given the performance equivalence on the pretraining task and the corrections of the algorithm to match the math of unpacked BERT, we are not expecting any negative impact on the downstream tasks. However, the review made us aware that the legends of the images need to be corrected. In fact, we showed that the validation accuracies are matching. Looking into downstream tasks and more importantly applying packing also to downstream tasks is an interesting area for potential future work, as we will mention in the paper.
>
>
>
> The sentence about "degradation of masked-language modelling accuracy on shorter sequences" was an anecdotal comment from early experiments (0.3% accuracy drop) and will be removed. In the original BERT implementation (and derivatives), per-token accuracy is calculated and not per-sequence. Hence, our approach for sentence correction is not required when packing. However, if each sentence gets the same weight in the final evaluation and the token accuracies do not get averaged, having 1 token incorrect in a short sequence with only 1 masked token compared to 1 token out of 100 masked tokens in a long sequence will make a significant difference. When we tested it, we get 0.3% less accuracy after training from a checkpoint with 300k samples. In short, accuracy evaluation and loss evaluation scheme should align. Since this is well known, we will remove this sentence.
>
>
>
> We agree that it would be good to validate if packing eliminates the need for the first phase in pre-training. However, in the RoBERTa paper, a simplified version of packing is already used to remove the first phase. Hence, it would be better to replace the sentence in our paper and mention the comparison to the RoBERTa paper.

---

> > ### Author Response · Authors · 2021-08-14
> > **Positional Embeddings**
> >
> > We added the following Sections to the paper to also include the positional embedding. Furthermore, we streamlined abstract, introduction, and conclusion to either refer to positional embeddings or instead refer to all the model changes collectively.
> >
> > **3.2    packedBERT: model changes**
> >
> > *This section describes how any vanilla BERT implementation should be modified for packed sequence processing, such that the behavior of the model is the same as when processing unpacked sequences. Preserving the mathematical equivalence is necessary to ensure existing BERT pre-training and fine-tuning practices remain valid, as well as being required by benchmarks such as MLPerf™[16].*
> >
> > **3.2.1    Positional embeddings for packed sequences**
> >
> > *The BERT model [5] uses three types of embeddings: token, segment, and positional embeddings.The latter is canonically implemented as a bias add operation, rather than a full embedding look-up.This is possible because the positional indices are the same for every sequence. However, when using the packed data format the position index needs to be reset with each new packed sequence.  For instance, when packing two sequences one of length 2 and one of length 3, the positional embedding indexes that need to be picked up are [0,1,0,1,2]. For achieve this, the bias add needs to be replaced by an embedding look-up to extract the correct positional embedding for each token in the pack. This also requires keeping an extra input which specifies the position of each token in its sequence.*
> >
> > Please let us know if this clarifies things sufficiently.

---

> > > ### Comment · Reviewer_Sn5a · 2021-09-01
> > > **Thanks for the detailed clarification**
> > >
> > > The author's response has addressed some of my concerns. Of course there are spaces for further improvements. I will change my overall rating to 6.

---

> > > > ### Author Response · Authors · 2021-09-02
> > > > **Thank you for the feedback and the changed rating**
> > > >
> > > > We will keep improving further. For example, we developed two animated videos that explain the packing algorithms much better.
> > > >
> > > > Also, we started looking into downstream tasks and training from scratch. Pretraining curves with sequence length 128 for phase 1 and 384 for phase 2 were quite close after some hyperparameter adjustment. When fine-tuning the pretrained model on SQuAD 1.1. (without packing for the fine-tuning part), we observed slightly higher performance with a model that was pretrained with packing compared to vanilla training. On the other hand, when switching from BERT Base to BERT Large, the vanilla training showed slightly better performance. After investigating the right choice of hyperparameter, we will add the findings to the appendix because the experiment confirms that packing keeps the performance the same while accelerating the processing significantly.

---

> > ### Author Response · Authors · 2021-08-19
> > **Figure for Section 3.3 and clarification on other "packing" approaches**
> >
> > We replaced the code for the loss adjustment in former Section 3.3  by a graphic that explains the process very well. Unfortunately, the openreview tool does not allow for sharing the graphic.
> >
> > Furthermore, we mention the packing of sentences in the BERT source code in our introduction now as well as the concatenation of sequences from different documents which results in reduced predictive quality (RoBERTa). Note, that we solve the efficient packing of sequences without contamination and perfect use of the provided sequence length in contrast to the concatenation of sentences or undesirable concatenation of sequences.

---

### Author Response · Authors · 2021-08-11
**Thank you for your review and helpful comments**

We thank all three reviewers for their helpful comments. The reviewers appreciated the well-designed experiments and analysis and the importance of the topic. Padding and packing are a crucial part of NLP but rarely published and discussed in detail. The reviews reveal that there seems to be some confusion in the state of the art on packing in NLP, which will require an extension of our introduction to clarify things especially in context of "packing" in the classical BERT algorithm, FasterTransformer or “sorted batched padding", and FULL-SENTENCE packing in RoBERTa. In contrast to our clarification of the state of the art, our three main contributions are not contested in the review and should be published.

First, to the best of our knowledge, we are the first to quantify padding inefficiencies across a variety of NLP datasets and point it out as a major concern.

Second, whereas some packing approaches exist in NLP, we are the first to address efficient packing, which could transfer to a variety of other applications even beyond NLP. Note that our approach gets close to perfect packing efficiency with minimal overhead in a noticeably brief period even for large scale datasets.

Third, we provide masking and loss-unpacking algorithms (together with hyperparameter adjustments) that can be applied to a variety of Transformers based tasks (like BERT and RoBERTa), to avoid unwanted contamination between disjunct documents or packed sequences without the need for custom kernels or hardware. For instance, the cross-contamination in RoBERTa is cited to hurt performance. We also showed that the overhead of our approach is minimal, compared to the achievable speed-up.

Luckily, the review also revealed that the explanation of how we handle the positional embedding was missing. We will gladly add it, as it is something we had already implemented for the experiments.

Our experiments that prove that our modifications do not significantly alter the convergence behavior vis-à-vis the original implementation are well received. Nonetheless, downstream tasks can be of interest but are not required to prove the validity of our approach. In fact, they would be at best tangential to the purpose of the experiments, which was to show the impact of packing not just on a single final accuracy value, but more so on the overall convergence behavior of the model.

Given these uncontested core contributions and the significant relevance for the community, we hope we can present the paper at NeurIPS after carefully addressing all review comments in the paper. Given our experience with the topic, we strongly believe that these additions are straightforward, and we already started modifying the paper, respectively.

---

### Decision · Program_Chairs · 2021-09-27

**Decision:**

Reject

**Comment:**

Packing is an extremely underrated technique for drastically improving NLP training times. In my experience (on datasets other than Wikipedia), packing typically speeds up convergence times by 10x. Therefore this paper adds visibility and acts as a user-guide to a technique that has largely been under-discussed or taken for granted in the sequence modeling community, perhaps because it is often treated as *just* an engineering trick. I thoroughly enjoyed reading the paper and the rigor it provides to this useful technique.

This paper provides two novel and well-optimized algorithms for packing that deserve merit. That said, QSu8 points out correctly that packing has been done in common frameworks for a while even if there has been no publication on it. Thus many of the ideas in this paper are not originated from this paper. For clarity, the paper should make these distinctions. These ideas include
1. The greedy packing algorithm outlined in QSu8's initial review and implemented in frameworks such as fairseq and lingvo
2. The handling of positional embeddings as discussed in the response to Sn5a
3. The masking of attention matrices to eliminate cross-contamination

I have carefully read the discussions and will summarize my remaining concerns here by way of (hopefully) constructive feedback. Basically I agree with Sn5a and QSu8 that more experimental comparisons are needed.
1. Ablation study on attention masking. I.e. what if you didn't have the attention mask and allowed for cross-contamination? There is a plausible reason for this to still work since the <s> token demarcates the different segments. It would be interesting to know how much quality is actually hindered.
2. Direct comparison with RoBERTa as suggested by QSu8.
3. Sorted batch padding. I concur with QSu8 that overfitting to the sequence lengths is not an issue in practice given the large datasets. I disagree with the move to disregard doing the comparison based on the idea that variable batch sizes are hard to implement  though. This technique is widely used in the lingvo sequence library (implemented here https://github.com/tensorflow/lingvo/blob/22fa0a004b568477b8814195c199cce2717f1f5b/lingvo/core/datasource.py#L608-L617, and an LM that defines the bucket sizes here https://github.com/tensorflow/lingvo/blob/master/lingvo/tasks/lm/params/one_billion_wds.py#L53-L54). For recurrent architectures, the variable batch size can significantly reduce the amount of padding.
4. Finally, while not required, I think experiments with packing on downstream tasks would strengthen the paper even more since that's where the speedup would be greatest  (>>2x) as pointed out by Sn5a.

As it stands, given the high standards of NeurIPS, I hesitate to give an accept until the additional comparisons above are conducted and incorporated into the paper. I am very optimistic about the direction of this paper and encourage authors to revise and resubmit to the next conference.